# Obesity and revision surgery, mortality, and patient-reported outcomes after primary knee replacement surgery in the National Joint Registry: A UK cohort study

**Jonathan Thomas Evans**[1‡]*, **Sofia Mouchti**[1‡], **Ashley William Blom**[1,2], **Jeremy Mark Wilkinson**[3], **Michael Richard Whitehouse**[1,2], **Andrew Beswick**[1], **Andrew Judge**[1,2,4]

1 Musculoskeletal Research Unit, Translational Health Sciences, Bristol Medical School, Bristol, United Kingdom, 2 National Institute for Health Research Bristol Biomedical Research Centre, University Hospitals Bristol NHS Foundation Trust and University of Bristol, Bristol, United Kingdom, 3 Department of Oncology and Metabolism, The Mellanby Centre for Bone Research, University of Sheffield, Metabolic Bone Unit, Northern General Hospital, Sheffield, United Kingdom, 4 Nuffield Department of Orthopaedics, Rheumatology and Musculoskeletal Sciences (NDORMS), University of Oxford, United Kingdom

‡ These authors share first authorship on this work.
* j.t.evans@bristol.ac.uk

**Data Availability Statement:** Data are available from the NJR research subcommittee researchers who meet the criteria for access to confidential

## Abstract

### Background

One in 10 people in the United Kingdom will need a total knee replacement (TKR) during their lifetime. Access to this life-changing operation has recently been restricted based on body mass index (BMI) due to belief that high BMI may lead to poorer outcomes. We investigated the associations between BMI and revision surgery, mortality, and pain/function using what we believe to be the world's largest joint replacement registry.

### Methods and findings

We analysed 493,710 TKRs in the National Joint Registry (NJR) for England, Wales, Northern Ireland, and the Isle of Man from 2005 to 2016 to investigate 90-day mortality and 10-year cumulative revision. Hospital Episodes Statistics (HES) and Patient Reported Outcome Measures (PROMs) databases were linked to the NJR to investigate change in Oxford Knee Score (OKS) 6 months postoperatively.

After adjustment for age, sex, American Society of Anaesthesiologists (ASA) grade, indication for operation, year of primary TKR, and fixation type, patients with high BMI were more likely to undergo revision surgery within 10 years compared to those with "normal" BMI (obese class II hazard ratio (HR) 1.21, 95% CI: 1.10, 1.32 ($p < 0.001$) and obese class III HR 1.13, 95% CI: 1.02, 1.26 ($p = 0.026$)). All BMI classes had revision estimates within the recognised 10-year benchmark of 5%. Overweight and obese class I patients had lower mortality than patients with "normal" BMI (HR 0.76, 95% CI: 0.65, 0.90 ($p = 0.001$) and HR 0.69, 95% CI: 0.58, 0.82 ($p < 0.001$)). All BMI categories saw absolute increases in OKS after 6 months (range 18–20 points). The relative improvement in OKS was lower in overweight

data. Access to the data used in this study can be requested via njrresearch@hqip.org.uk. Full details of how to request NJR data for research can be found at: http://www.njrcentre.org.uk/njrcentre/Research/Research-requests.

**Funding:** JTE, SM, AWB, MRW, AB and AJ were supported by the NIHR Biomedical Research Centre at University Hospitals Bristol and Weston NHS Foundation Trust and the University of Bristol. The funders had no role in study design, data collection and analysis, decision to publish, or preparation of the manuscript.

**Competing interests:** I have read the journal's policy and the authors of this manuscript have the following competing interests: AJ receives consultancy fees from Freshfields, Bruckhaus, Derringer and is a paid member of data safety and monitoring board for Anthera Pharmecuticals Ltd. MRW has received a research grant (co-applicant) from Stryker for an unrelated knee replacement trial, MRW's institution has been paid for work he has done developing and delivering educational presentations on hip replacement and Basic Sciences for DePuy and Heraeus.

**Abbreviations:** AIC, Akaike information criterion; ASA, American Society of Anaesthesiologists; BIC, Bayes information criterion; BMI, body mass index; HES, Hospital Episodes Statistics; HR, hazard ratio; IMD, Index of Multiple Deprivation; MDC, minimal detectable change; NHS, National Health Service; NJR, National Joint Registry; OKS, Oxford Knee Score; PROMs, Patient Reported Outcome Measures; RECORD, Reporting of studies Conducted using Observational Routinely-collected Data; TKR, total knee replacement; WHO, World Health Organization.

and obese patients than those with "normal" BMI, but the difference was below the minimal detectable change (MDC; 4 points). The main limitations were missing BMI particularly in the early years of data collection and a potential selection bias effect of surgeons selecting the fitter patients with raised BMI for surgery.

## Conclusions

Given revision estimates in all BMI groups below the recognised threshold, no evidence of increased mortality, and difference in change in OKS below the MDC, this large national registry shows no evidence of poorer outcomes in patients with high BMI. This study does not support rationing of TKR based on increased BMI.

## Author summary

### Why was this study done?

- While total knee replacements (TKRs) are generally considered safe and effective, it has been suggested that patients with high body mass index (BMI) are at increased risk of poor outcomes, leading to policies restricting who is referred for surgery.
- Previous studies of the impact of BMI have used smaller datasets or have focused on a single outcome rather than the wider focus of this article, which includes mortality, implant survival, and patient-reported outcomes.
- We aimed to investigate whether patients with a raised BMI operated on within the National Joint Registry (NJR) had demonstrably worse outcomes following TKR.

### What did the researchers do and find?

- We analysed 493,710 TKRs implanted between 2005 and 2016 to investigate the proportion of patients that died within 90 days, how many implants needed revising (redo surgery) after 10 years, and the changes between preoperative and 6-month postoperative Oxford Knee Score (OKS).
- Patients with raised BMI (according to the World Health Organization (WHO) categories) were compared to those with a "normal" BMI.
- Patients in the "overweight" and "obese" groups had a lower 90-day mortality than those with "normal" BMI.
- TKR in patients with raised BMI were more likely to have been revised after 10 years, although the cumulative revision estimate in all groups was below the benchmark of 5% generally considered to be acceptable.
- All patient groups demonstrated an improvement in OKS after 6 months. The "overweight" and "obese" groups demonstrated a smaller relative improvement compared to the "normal" group; however, this relative difference was below the threshold considered to be clinically meaningful.

## What do these findings mean?

- There does not appear to be any evidence to support clinically relevant worse outcomes following TKR for patients with a raised BMI in the NJR between 2005 and 2016.

- These findings do not support restriction of referral for knee replacement based on BMI alone. It appears that even if some patients with raised BMI are at risk of poorer outcomes, the outcomes remain acceptable by contemporary standards, and the selection process of orthopaedic surgeons is effective at identifying the correct patients to operate on at a population level.

## Introduction

Total knee replacement (TKR) is one of the most common orthopaedic operations and is generally considered to be both safe, cost-effective, and clinically effective in reducing symptoms of pain and functional limitation in most patients [1,2]. Almost 1 in 10 people in the UK can expect to receive a TKR at some point in their lifetime, and approximately 100,000 have been performed in the UK each year for the last 4 years [3–5]. The main reasons for performing a TKR are joint pain and/or functional limitation in combination with radiographic evidence of arthritis; despite this, there is no consensus on the severity of symptoms that indicate the need for surgery [2,6,7]. Performing TKRs on the wrong patients may lead to poorer outcomes and lead to early revision surgery, which is both less effective than primary surgery and costly to patients and the health service [8,9]. Specific risk factors for poor outcomes that have previously been described include greater age, comorbidities, frailty, high body mass index (BMI), psychological factors, and the patient having a poor expectation of the success of surgery [10–13]. With an ageing population, the number of people having a TKR can be expected to increase, placing an increasing burden on the National Health Service (NHS) in respect of funding and capacity [14].

There is growing evidence that some commissioners of health services in the UK are either restricting access to TKR for patients with high BMI or encouraging weight loss prior to referral for surgery [15,16]. This may be as a result of a belief that these patients are at a higher risk of complications. Surgeons may express concerns that increased load on a prosthesis increases the risk of failure due to loosening or wear or that the operation itself is more difficult, resulting in an increase in perioperative problems [17]. This is despite evidence that overall, the absolute risk of postoperative complications within the first 6 months of TKR is low in patients with a high BMI [18].

National guidance in the UK is clear that in patients with clinical osteoarthritis, while interventions to achieve weight loss are recommended, a high BMI and other patient specific factors should not be barriers to referral for joint replacement [6]. In contrast to this, there is some evidence from joint registries, observational cohort studies, and routine hospital admission data that high BMI is associated with poorer outcomes with regard to pain and function, mortality, complications, and need for revision surgery [18,19]. Whether these observed associations transfer to be clinically meaningful is as yet unclear.

Using data from the National Joint Registry (NJR) for England, Wales, Northern Ireland, and the Isle of Man, our aim is to describe the association of BMI at the time of surgery with revision after 10 years, 90-day mortality, and patient-reported outcomes 6 months following

primary TKR and to consider the clinical importance of any observed association. This is of importance for both future commissioning and clinical decision-making.

## Methods

### Study design and data source

We performed an observational cohort study using data obtained from the NJR. Since April 2009, Patient Reported Outcome Measures (PROMs) data have been collected on TKRs performed in public hospitals in England, most notably for this study, preoperative and 6-month postoperative Oxford Knee Scores (OKSs) [20].

### Data linkages, participants, and inclusion criteria

The NJR started collecting BMI data on April 1, 2005, and we investigated patients undergoing primary TKR from this date up to and including December 31, 2016 for revision and mortality outcomes. Data were excluded on patients with missing or implausible BMI, age or sex, unspecified TKR fixation type, TKRs performed for trauma as well as for patients without a specified NHS number (preventing linkage) or with an unknown indication. Linkage between PROMs and the NJR was made via the Hospital Episodes Statistics (HES) database, which records details of all hospital admissions in England using the same exclusion criteria. HES data and subsequently PROMs data were only available up to December 30, 2014.

### Outcomes

The outcome variables for this study are revision surgery (defined as the addition, removal, or modification of any part of the construct) [3], mortality within 90 days of the primary operation, and patient-reported outcome assessed using the change in OKS after 6 months. The OKS is a patient-completed questionnaire that assesses knee pain and function with 12 questions, each scored from 0 to 4, completed using Likert scales, and the scores are summed to give a score from 0 (worst) to 48 (best) [20]. In cohort studies (such as the NJR), the minimal detectable change (MDC) in OKS at the group level has been shown to be 4 points [21].

### Exposure variable

The primary exposure of interest is BMI at the time of operation defined according to the World Health Organization (WHO) International Classification: $<18.5$ kg/m$^2$ (underweight); 18.5 to 24.99 kg/m$^2$ (normal weight); 25 to 29.99 kg/m$^2$ (overweight); 30 to 34.99 kg/m$^2$ (obese class I); 35 to 39.99 kg/m$^2$ (obese class II); and $>40$ kg/m$^2$ (obese class III).

### Confounding variables

Confounding variables considered included age at primary TKR grouped as $<50$, 50 to 54, 55 to 59, 60 to 64, 65 to 69, 70 to 74, 75 to 79, 80 to 84, and $\geq 85$ measured in years; sex; American Society of Anaesthesiologists (ASA) physical status classification grouped as P1, P2, P3, or P4 to P5; year of receiving the primary TKR grouped as 2005 to 2007 and as individual years between 2008 and 2016; cemented, uncemented, or hybrid fixation; reason for operation classified as osteoarthritis, osteoarthritis plus another indication, or other indications only; quintiles of the Index of Multiple Deprivation (IMD) coded between 1 (most deprived) and 5 (least deprived); Charlson comorbidity index grouped as 0 (no comorbidities), 1 (mild), 2 (moderate), and 3+ (severe) comorbidities; and preoperative EQ5D 3L Anxiety/Depression domain. The IMD is the official measure of relative deprivation for small areas (Lower Layer Super Output Areas) in England. The measure is calculated using 7 domains including income,

employment, education, health, crime, and environment. It ranks every small area from 1 (most deprived) to 32,844 (least deprived) [22].

## Statistical analysis

We plotted Kaplan–Meier estimates with risk tables to explore cumulative probability of revision up to 11 years and death up to 90 days for the BMI categories. Time zero was considered the time of the primary operation, patients were considered to have exited the study after the first revision episode was observed, and patients were censored upon death and administratively censored on December 31, 2016.

We used flexible parametric survival models as described by Royston and Parmar to investigate the association between BMI category and the risk of revision [23]. To choose a suitable scale and baseline complexity for the model, we fitted a univariable model (on the BMI category). We assessed choice of scale and number of knots for baseline spline function by inspecting the Akaike information criterion (AIC) and Bayes information criterion (BIC) statistics. We used Cox proportional hazards regression models to investigate 90-day mortality. We adjusted for age, sex, ASA grade, indication for operation, and year of primary TKR. The assumption of proportionality of hazards was assessed visually and through the use of Schoenfeld residuals.

Linear regression modelling (ANCOVA) is used to describe the association of BMI on 6-month OKS, adjusting for preoperative OKS as a covariate in the model and known available confounders. As there was evidence of heteroscedasticity (variance of the residuals is nonconstant), robust standard errors were used with the sandwich variance estimator [24]. Stata 14.2 was used for all analyses (Stata Statistical Software: Release 14, Stata, College Station, Texas, United States of America).

For survival outcomes, each knee replacement was treated as an individual; this is possible given the nature of reporting of both primary and revisions in the NJR. For PROMs and mortality analyses, however, same-day knee replacements could not be interpreted individually. For this reason, in same-day TKRs, only 1 was selected at random to contribute to the analyses to avoid duplication of data.

## Sensitivity analysis

We further adjusted for confounders that can be derived only from the subset of patients with linked HES data (Charlson comorbidity score and IMD deprivation score) to estimate revision and mortality. In the PROMs analysis, this included further adjustment for the preoperative EQ5D 3L Anxiety/Depression question score. In response to peer review, all models for primary outcomes were run with BMI as a continuous variable using restricted cubic splines with knots at cutoffs of WHO categories.

## Missing data

A comparison of demographic characteristics of participants with and without a recorded BMI was conducted to investigate the potential for selection bias.

## Planning of analyses

The analysis plan was made prior to the start of all analyses and agreed among coauthors. No data-driven changes to the analysis plan were made. An additional sensitivity analysis with BMI as a continuous variable using splines (at WHO cutoffs) to investigate nonlinearity was included in response to peer review.

Reporting of the study was in keeping with guidance provided in the Reporting of studies Conducted using Observational Routinely-collected Data (RECORD) statement (S1 Checklist) [25].

Approval for this study was granted by the NJR research subcommittee reference. Written consent was granted by patients for inclusion of their data and its use in research within the NJR for England, Wales, Northern Ireland, and the Isle of Man.

## Results

### Participants

After exclusions, 493,710 TKRs remained to investigate revisions and 90-day mortality (Fig 1), with a maximum follow-up time of 11 years and a mean of 3.8 years. This dataset accounted for 56% of the total number of primary TKRs recorded in the NJR to December 31, 2016.

In linked PROMs, HES, and NJR datasets, 237,288 primary TKR operations were performed between March 26, 2009 and December 30, 2014 (S1 Fig). After applying the exclusion criteria, 165,193 primary TKR were available to investigate the association of BMI with the OKS patient-reported outcome.

### Descriptive data

Overall, 57% of operations included in the NJR between 2005 and 2016 had BMI recorded. Completeness of overall BMI data in the NJR has improved over time; in 2005, of the 31,733 operations, 17.0% had BMI data, compared to 79.5% of the 88,078 operations in 2016. Demographics were similar between the 2 datasets with either complete or incomplete BMI data (Table 1).

Patient characteristics in different BMI categories are summarised in Table 2. Overall, 55.4% of patients were obese (BMI $\geq$30 kg/m$^2$), and 0.3% were underweight (BMI <18.5 kg/m$^2$). Low ASA grades were more frequently observed in people with BMI <35 kg/m$^2$ (WHO obese class I or below), while higher ASA grades were more common in underweight or obese class II and III patients (BMI <18.5 and $\geq$35 kg/m$^2$). The majority (>95%) of TKRs were cemented in all BMI categories.

### Revision

Fig 2 demonstrates that the cumulative probability of revision rises with increasing BMI at the time of operation. Table 3 shows the number of knee replacements "at risk" (not yet failed or censored for death or administratively) at each time point for each BMI class in the original dataset, from which the model was built. After 10 years, patients with BMI $\geq$40 kg/m$^2$ had 4.0% (95% CI: 3.6, 4.5) cumulative probability of revision compared with 2.8% (95% CI: 2.5, 3.3) in those with BMI 18.5 to 24.99 kg/m$^2$ (Table 4). Table 5 presents the hazard ratios (HRs) for each BMI group (derived from the flexible parametric models) for revision relative to patients with BMI of 18.5 to 24.99 kg/m$^2$ encompassing the full 11 years of follow-up. The adjusted model shows that patients with BMI 30 to 34.99 kg/m$^2$, 35 to 39.99 kg/m$^2$, and $\geq$40 kg/m$^2$ were 8% (HR 1.08, 95% CI: 0.99, 1.18 ($p = 0.073$)), 21% (HR 1.21, 95% CI: 1.10, 1.32 ($p < 0.001$)), and 13% (HR 1.13, 95% CI: 1.02, 1.26 ($p = 0.026$)) more likely to undergo a revision than patients with BMI 18.5 to 24.99 kg/m$^2$, respectively, although it should be noted that the confidence intervals for the 30 to 34.99 kg/m$^2$ category do cross the null value. Fig 3 shows the hazard of revision when BMI is modelled as a continuous variable with splines at WHO cutoffs. This model is consistent with models using BMI as a categorical variable.

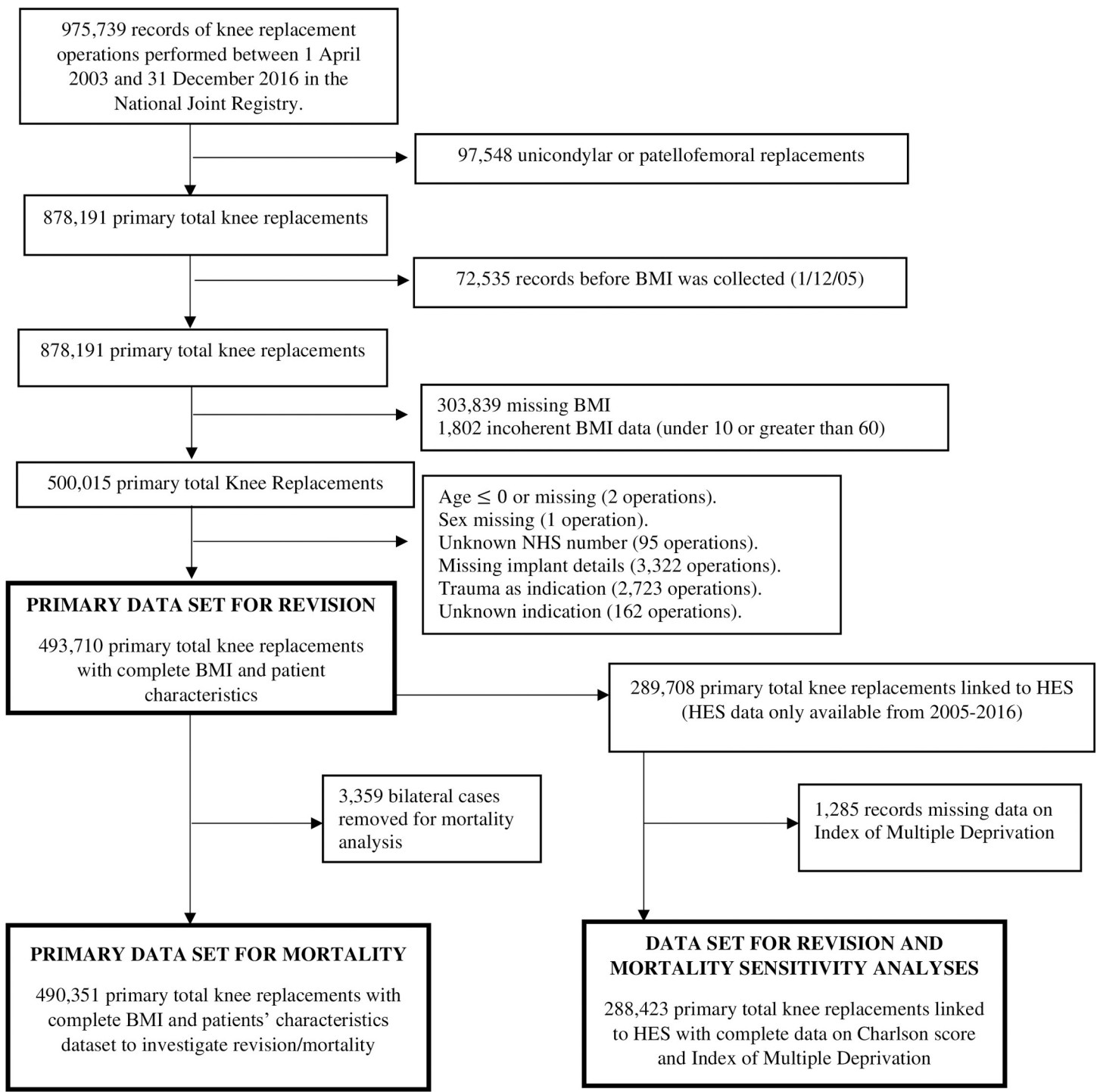

**Fig 1. Flow diagram showing the availability of mortality and revision data after primary TKR.** BMI, body mass index; HES, Hospital Episodes Statistics; NHS, National Health Service; TKR, total knee replacement.

## Mortality

Table 6 shows that patients with BMI 25 to 29.99 kg/m² and 30 to 34.99 kg/m² had 24% (HR 0.76, 95% CI: 0.65, 0.90 ($p = 0.001$)) and 31% (HR 0.69, 95% CI: 0.58, 0.82 ($p < 0.001$)) lower

**Table 1. Distribution of sex, ASA grade, fixation type, and age in datasets with complete and incomplete BMI records.**

| | | Complete (N = 493,710) | | Incomplete (N = 384,481) | |
|---|---|---|---|---|---|
| | | N | % | N | % |
| Sex | Female | 283,161 | 57.4 | 221,450 | 57.6 |
| | Male | 210,549 | 42.6 | 163,030 | 42.4 |
| | Missing | 0 | 0 | 1 | 0 |
| ASA grade | P1 | 48,134 | 9.75 | 51,405 | 13.4 |
| | P2 | 362,745 | 73.5 | 272,432 | 70.9 |
| | P3 | 81,342 | 16.5 | 58,931 | 15.3 |
| | P4–P5 | 1,489 | 0.3 | 1,713 | 0.45 |
| Fixation type | Cemented | 473,303 | 95.9 | 355,270 | 92.4 |
| | Uncemented | 17,380 | 3.52 | 23,340 | 6.07 |
| | Hybrid | 3,027 | 0.61 | 5,871 | 1.53 |
| Age in years | <50 | 9,883 | 2 | 8,268 | 2.15 |
| | 50–54 | 20,024 | 4.06 | 14,131 | 3.68 |
| | 55–59 | 40,688 | 8.24 | 31,392 | 8.16 |
| | 60–64 | 72,014 | 14.6 | 54,850 | 14.3 |
| | 65–69 | 96,459 | 19.5 | 71,053 | 18.5 |
| | 70–74 | 98,844 | 20 | 77,452 | 20.1 |
| | 75–79 | 85,619 | 17.3 | 70,086 | 18.2 |
| | 80–84 | 50,293 | 10.2 | 40,999 | 10.7 |
| | ≥85 | 19,886 | 4.03 | 16,250 | 4.23 |

ASA, American Society of Anaesthesiologists; BMI, body mass index.

90-day mortality rate than patients with a normal BMI. Fig 4 demonstrates the mortality sensitivity analysis of the Cox model with BMI modelled as a continuous variable and is consistent with the findings form the model with BMI as a categorical variable.

## Oxford Knee Score

The crude increase in OKS between pre- and 6-month postoperative assessments was similar across all BMI groups (range 18 to 20 points) and well above the minimal important change of

**Table 2. Patient characteristics for sex, age, ASA grade, and fixation type by BMI category.**

| | | <18.5 kg/m$^2$ | 18.5–24.99 kg/m$^2$ | 25–29.99 kg/m$^2$ | 30–34.99 kg/m$^2$ | 35–39.99 kg/m$^2$ | ≥40 kg/m$^2$ |
|---|---|---|---|---|---|---|---|
| n (%) | | 1,338 (0.27) | 49,860 (10.10) | 168,947 (34.22) | 159,056 (32.22) | 80,166 (16.24) | 34,343 (6.96) |
| Sex n (%) | Female | 1,025 (76.61) | 30,666 (61.50) | 85,150 (50.40) | 87,863 (55.24) | 52,759 (65.81) | 25,698 (74.83) |
| | Male | 313 (23.39) | 19,194 (38.50) | 83,797 (49.60) | 71,193 (44.76) | 27,407 (34.19) | 8,645 (25.17) |
| Age median (IQR) | Female | 74 (66, 80) | 74 (67, 80) | 73 (66, 78) | 70 (64, 76) | 67 (61, 73) | 64 (58, 70) |
| | Male | 70 (63, 78) | 74 (67, 80) | 71 (65, 77) | 69 (63, 74) | 66 (61, 72) | 64 (59, 69) |
| ASA grade n (%) | P1 | 109 (8.15) | 6,734 (13.51) | 21,105 (12.49) | 14,719 (9.25) | 4,443 (5.54) | 1,024 (2.98) |
| | P2 | 904 (67.56) | 35,812 (71.83) | 125,847 (74.49) | 120,151 (75.54) | 59,200 (73.85) | 20,831 (60.66) |
| | P3 | 317 (23.69) | 7,145 (14.33) | 21,618 (12.80) | 23,806 (14.97) | 16,265 (20.29) | 12,191 (35.50) |
| | P4–P5 | 8 (0.60) | 169 (0.34) | 377 (0.22) | 380 (0.24) | 258 (0.32) | 297 (0.86) |
| Fixation type n (%) | Cemented | 1,306 (97.61) | 47,889 (96.05) | 161,854 (95.80) | 152,224 (95.70) | 76,958 (96.00) | 33,072 (96.30) |
| | Uncemented | 22 (1.64) | 1,640 (3.29) | 6,115 (3.62) | 5,806 (3.65) | 2,752 (3.43) | 1,045 (3.04) |
| | Hybrid | 10 (0.75) | 331 (0.66) | 978 (0.58) | 1,026 (0.65) | 456 (0.57) | 226 (0.66) |

ASA, American Society of Anaesthesiologists; BMI, body mass index.

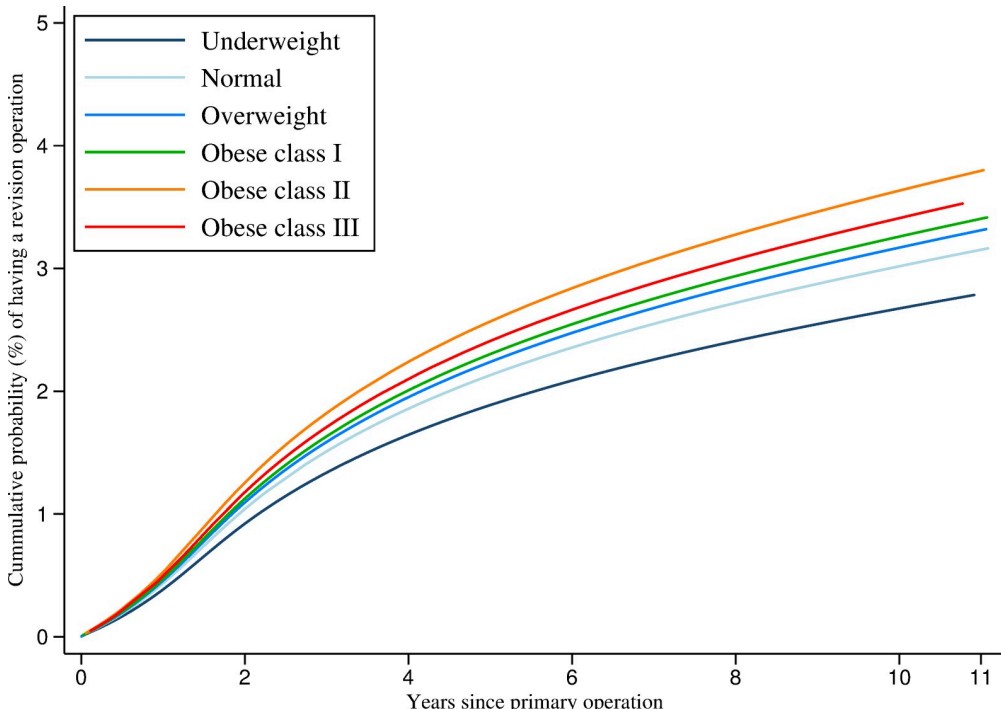

**Fig 2. Flexible parametric model estimates of cumulative probability of revision up to 11 years after primary TKR by BMI category.** BMI, body mass index; TKR, total knee replacement.

4/48 reported by Beard and colleagues (Table 4) [21]. After adjusting for age, sex, ASA, indication, fixation, year of operation, and anxiety status, the relative increase in OKS (between preoperative and 6-month postoperative) for patients with raised BMI was smaller relative to patients with a "normal" BMI (Table 7). Fig 5 shows the same model with BMI as a continuous variable using splines at WHO cutoffs.

Fig 6 illustrates the change between the pre- and postoperative OKS across the BMI categories. It highlights the substantial absolute change in OKS across all BMI categories compared to the small relative differences in the postoperative OKS between BMI categories.

## Sensitivity analysis

Further analyses adjusting for additional confounders of deprivation and Charlson comorbidity did not change the findings with effect sizes being similar (S3 Table).

**Table 3. Numbers of knee replacements at risk at specified time points in the dataset from which the model was built.**

| | Years since primary operation | | | | | | | |
|---|---|---|---|---|---|---|---|---|
| | **0** | **3** | **5** | **7** | **8** | **9** | **10** | **11** |
| Underweight | 1,338 | 807 | 505 | 236 | 153 | 66 | 20 | 0 |
| Normal | 49,860 | 28,400 | 17,064 | 8,395 | 8,395 | 2,085 | 199 | 36 |
| Overweight | 168,947 | 95,567 | 57,276 | 27,243 | 27,243 | 6,029 | 2,275 | 115 |
| Obese class I | 159,056 | 88,937 | 52,401 | 24,622 | 24,622 | 5,019 | 1,878 | 106 |
| Obese class II | 80,166 | 43,631 | 25,254 | 11,343 | 11,343 | 2,157 | 758 | 52 |
| Obese class III | 343,433 | 18,672 | 10,728 | 4,752 | 4,752 | 917 | 336 | 21 |

**Table 4. Median and IQR of the pre- and postoperative OKS, cumulative percentage probability (KM estimates) of revision with 95% CI at 3, 5, 7, and 10 years, and cumulative percentage probability of mortality after 90 days (KM estimates) with 95% CI at 30, 60, and 90 days by BMI category.**

| | | $<$18.5 kg/m$^2$ | 18.5–24.99 kg/m$^2$ | 25–29.99 kg/m$^2$ | 30–34.99 kg/m$^2$ | 35–39.99 kg/m$^2$ | $\geq$40 kg/m$^2$ |
|---|---|---|---|---|---|---|---|
| OKS median (IQR) | | | | | | | |
| Preoperative | | 16 (10, 23) ($n$ = 386) | 20 (14, 26) ($n$ = 15,319) | 20 (14, 25) ($n$ = 55,001) | 18 (13, 24) ($n$ = 53,496) | 16 (11, 21) ($n$ = 27,498) | 14 (9, 19) ($n$ = 11,608) |
| Postoperative | | 36 (28, 42) ($n$ = 293) | 39 (31, 44) ($n$ = 12,807) | 38 (31, 44) ($n$ = 46,927) | 37 (29, 42) ($n$ = 44,549) | 35 (26, 41) ($n$ = 22,176) | 33 (24, 40) ($n$ = 8,977) |
| Cumulative probability of revision (95% CI) | | | | | | | |
| Years since primary TKR | 3 | 1.14 (0.65, 2.01) | 1.24 (1.14, 1.36) | 1.38 (1.32, 1.45) | 1.59 (1.52, 1.66) | 1.95 (1.84, 2.06) | 2.06 (1.89, 2.24) |
| | 5 | 1.77 (1.07, 2.93) | 1.70 (1.57, 1.85) | 1.95 (1.87, 2.04) | 2.21 (2.12, 2.30) | 2.74 (2.60, 2.88) | 2.87 (2.65, 3.10) |
| | 7 | 2.29 (1.39, 3.77) | 2.10 (1.92, 2.28) | 2.40 (2.30, 2.51) | 2.68 (2.57, 2.79) | 3.26 (3.09, 3.44) | 3.49 (3.21, 3.79) |
| | 10 | 2.29 (1.39, 3.77) | 2.83 (2.46, 3.26) | 2.91 (2.74, 3.09) | 3.27 (3.08, 3.47) | 3.79 (3.50, 4.10) | 4.02 (3.62, 4.47) |
| Cumulative probability of mortality (95% CI) | | | | | | | |
| Days since primary TKR | 30 | 0.38 (0.16, 0.90) | 0.24 (0.21, 0.29) | 0.16 (0.14, 0.18) | 0.11 (0.10, 0.13) | 0.11 (0.09, 0.14) | 0.15 (0.11, 0.19) |
| | 60 | 0.68 (0.36, 1.31) | 0.34 (0.30, 0.40) | 0.23 (0.21, 0.25) | 0.16 (0.14, 0.18) | 0.17 (0.14, 0.20) | 0.20 (0.16, 0.25) |
| | 90 | 0.76 (0.41, 1.41) | 0.46 (0.41, 0.53) | 0.29 (0.27, 0.32) | 0.21 (0.19, 0.23) | 0.21 (0.18, 0.25) | 0.24 (0.19, 0.29) |

BMI, body mass index; KM, Kaplan–Meier; OKS, Oxford Knee Score; TKR, total knee replacement.

## Discussion

### Statement of principal findings

In this study using a large national joint replacement registry, after adjusting for age, sex, ASA, indication for operation, year of operation, and fixation type, patients classified as overweight or obese (BMI $\geq$25kg/m$^2$) had a reduced 90-day mortality risk but an increased risk of revision surgery compared to those in the "normal" category. The 10-year cumulative risk of revision in patients with BMI 18.5 to 24 kg/m$^2$ (reference group) was 2.8% and ranged from 2.3% in people with lowest BMI to 4.0% in those with the highest BMI. Patients in the "underweight" group (BMI $<$18.5kg/m$^2$) had the highest mortality 90 days after TKR, but even in this large national arthroplasty registry dataset, the number of patients affected was small with 10 deaths in 1,338 patients. Regarding PROMs, all categories of BMI showed an absolute improvement in median OKS after 6 months compared to median preoperative scores. The relative improvement in OKS was slightly lower in overweight and obese patients at the time of surgery

**Table 5. HR, 95% CI, and $p$-value for coefficients of BMI categories extracted from the flexible parametric models to investigate the association of BMI with revision after primary TKR.**

| | Unadjusted model | | | Adjusted model | | |
|---|---|---|---|---|---|---|
| | HR | 95% CI | $p$-value | HR | 95% CI | $p$-value |
| $<$18.5 kg/m$^2$ | 0.96 | (0.60, 1.54) | 0.872 | 0.88 | (0.55, 1.41) | 0.608 |
| 18.5–24.99 kg/m$^2$ (reference) | 1.00 | | | 1.00 | | |
| 25–29.99 kg/m$^2$ | 1.12 | (1.03, 1.22) | 0.007 | 1.05 | (0.97, 1.14) | 0.252 |
| 30–34.99 kg/m$^2$ | 1.26 | (1.16, 1.37) | $<$0.001 | 1.08 | (0.99, 1.18) | 0.073 |
| 35–39.99 kg/m$^2$ | 1.54 | (1.41, 1.68) | $<$0.001 | 1.21 | (1.10, 1.32) | $<$0.001 |
| $\geq$40 kg/m$^2$ | 1.64 | (1.48, 1.82) | $<$0.001 | 1.13 | (1.02, 1.26) | 0.026 |

Adjusted model adjusts for age, sex, ASA grade, indication for operation, year of primary TKR, and fixation type. Both models were fitted on the hazard scale with 4 degrees of freedom.

ASA, American Society of Anaesthesiologists; BMI, body mass index; HR, hazard ratio; TKR, total knee replacement.

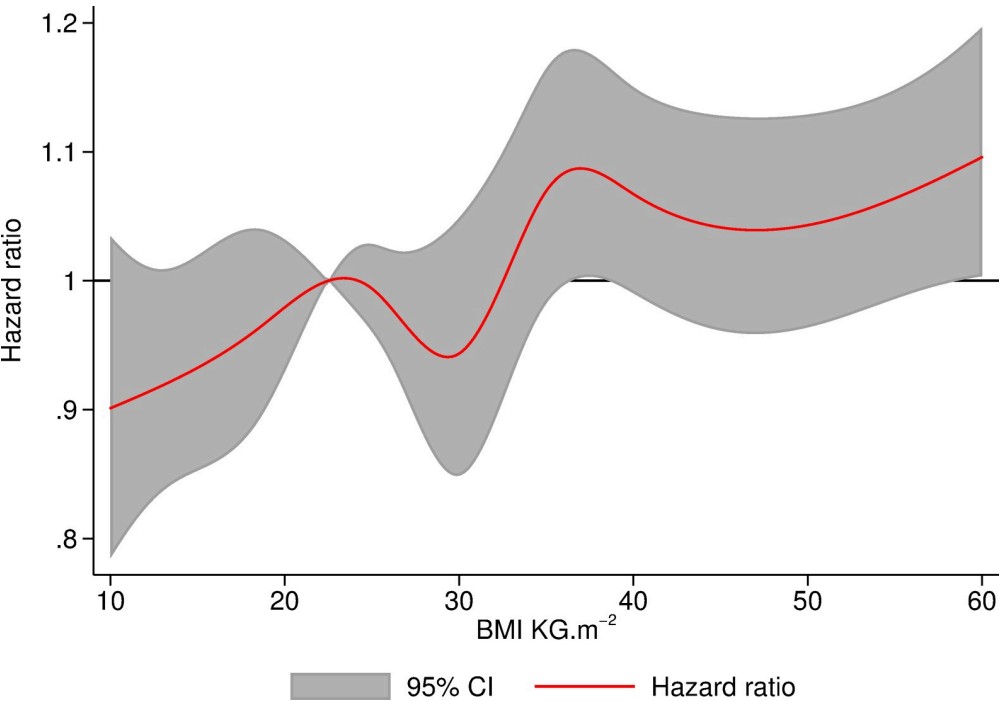

**Fig 3. Hazard of revision within 11 years of TKR relative to patients with BMI of 22.5 modelled using flexible parametric survival analysis using BMI as a continuous variable with restricted cubic splines at cutoffs of WHO criteria.** BMI, body mass index; TKR, total knee replacement; WHO, World Health Organization.

compared to patients with "normal" BMI, and the differences between groups were below the minimally important difference in change score. The 6-month absolute OKS appeared lower in higher BMI categories relatively, which reflects a lower starting point in these categories.

## Strengths and weaknesses of the study

To our knowledge, this is the first study on obesity and knee replacement to examine all 3 domains of implant revision, mortality, and patient-reported outcomes. The failings of examining single domains have previously been highlighted, in that just because a TKR has not been revised does not necessarily mean it was a success [26]. We used what we believe is the

**Table 6. HR, 95% CI, and *p*-value for coefficients of BMI categories extracted by Cox proportional hazards models to investigate the association of BMI with mortality within 90 days of primary TKR.**

| | Unadjusted model | | | Adjusted model | | |
|---|---|---|---|---|---|---|
| | HR | 95% CI | *p*-value | HR | 95% CI | *p*-value |
| <18.5 kg/m$^2$ | 1.65 | (0.87, 3.10) | 0.122 | 1.64 | (0.87, 3.09) | 0.128 |
| 18.5–24.99 kg/m$^2$ (reference) | 1.00 | | | 1.00 | | |
| 25–29.99 kg/m$^2$ | 0.64 | (0.55, 0.75) | <0.001 | 0.76 | (0.65, 0.90) | 0.001 |
| 30–34.99 kg/m$^2$ | 0.46 | (0.39, 0.55) | <0.001 | 0.69 | (0.58, 0.82) | <0.001 |
| 35–39.99 kg/m$^2$ | 0.46 | (0.38, 0.56) | <0.001 | 0.88 | (0.72, 1.09) | 0.247 |
| ≥40 kg/m$^2$ | 0.51 | (0.40, 0.66) | <0.001 | 1.17 | (0.90, 1.54) | 0.247 |

Adjusted model adjusts for age, sex, ASA grade, indication for operation, and year of primary TKR.

ASA, American Society of Anaesthesiologists; BMI, body mass index; HR, hazard ratio; TKR, total knee replacement.

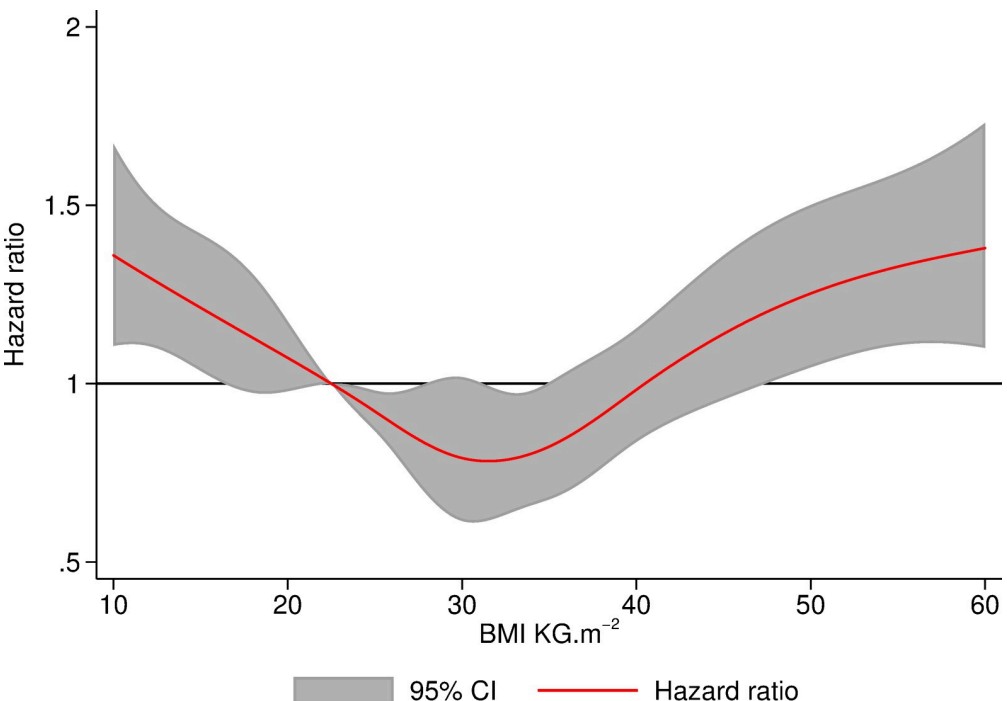

**Fig 4. Hazard of death within 90 days of TKR relative to patients with BMI of 22.5 modelled using Cox proportional hazards using BMI as a continuous variable with restricted cubic splines at cutoffs of WHO criteria.** BMI, body mass index; TKR, total knee replacement; WHO, World Health Organization.

largest joint replacement registry in the world, with near complete coverage of all operations performed in the target population. Analyses were not restricted to certain groups of patients or implant providers, allowing us to generalise the results to most patients undergoing elective primary TKR in England and Wales. The most notable limitation is the missing data on BMI. Before 2005, this variable was not collected, and between 2005 and 2016, the completeness of BMI data in our study dataset rose from 20.5% to 83.0%. Patient demographics were similar between operations with complete and non-complete BMIs, suggesting that there was unlikely to be responder bias. The main differences between groups (Table 1) were the distribution of patients between the ASA 1 and 2 groups and fixation type. Results of patients with ASA 1 and 2 tend to be similar, and so we do not feel this is likely to have biased results. More patients

**Table 7. Estimates of BMI category coefficients to predict the mean increase or decrease in postoperative OKS.**

| | Unadjusted model | | | Adjusted model | | |
|---|---|---|---|---|---|---|
| | Coefficient | 95% CI | *p*-value | Coefficient | 95% CI | *p*-value |
| <18.5 kg/m$^2$ | −1.04 | (−2.08, −0.01) | 0.044 | −0.74 | (−1.75, 0.27) | 0.150 |
| 18.5–24.99 kg/m$^2$ (reference) | 0.00 | | | 0.00 | | |
| 25–29.99 kg/m$^2$ | −0.24 | (−0.42, −0.07) | 0.005 | −0.35 | (−0.52, −0.18) | 0.001 |
| 30–34.99 kg/m$^2$ | −1.07 | (−1.24, −0.89) | <0.001 | −1.10 | (−1.27, −0.92) | <0.001 |
| 35–39.99 kg/m$^2$ | −1.96 | (−2.16, −1.76) | <0.001 | −1.82 | (−2.02, −1.61) | <0.001 |
| ≥40 kg/m$^2$ | −2.83 | (−3.07, −2.58) | <0.001 | −2.20 | (−2.46, −1.93) | <0.001 |

Adjusted model adjusts for age, sex, ASA grade, indication for operation, fixation type, year of primary TKR, and anxiety status.

ASA, American Society of Anaesthesiologists; BMI, body mass index; OKS, Oxford Knee Score; TKR, total knee replacement.

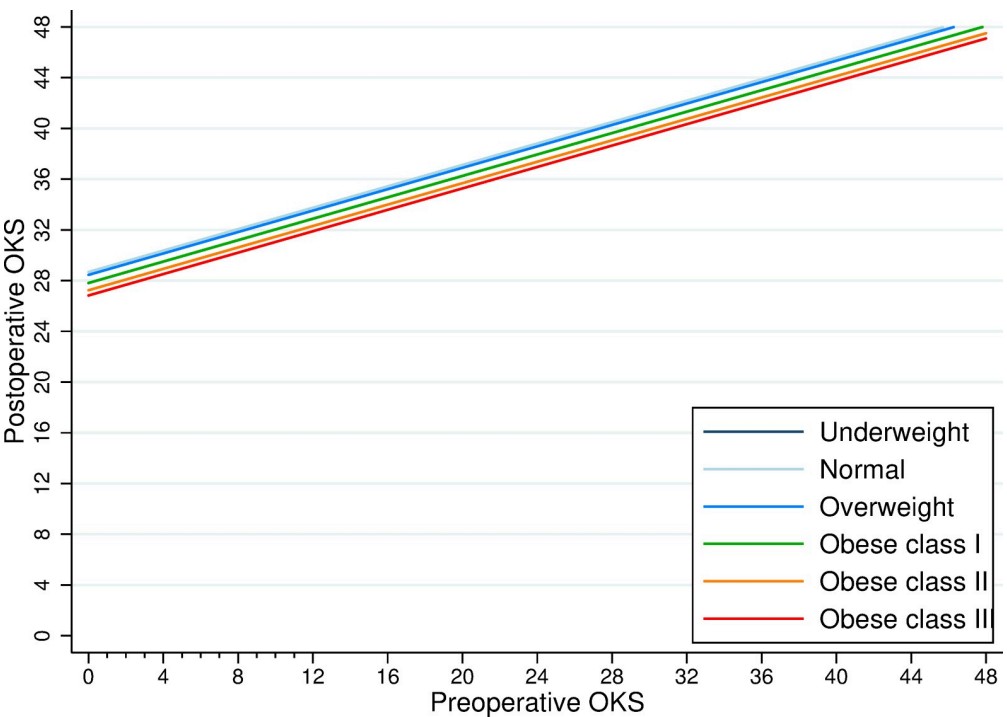

**Fig 5. Change in mean OKS 6 months after TKR relative to patients with a BMI of 22.5 modelled using linear regression using BMI as a continuous variable with restricted cubic splines at cutoffs of WHO criteria.** Model adjusting for age, sex, ASA grade, indication for operation, fixation type, year of primary TKR, and anxiety status. ASA, American Society of Anaesthesiologists; BMI, body mass index; OKS, Oxford Knee Score; TKR, total knee replacement; WHO, World Health Organization.

with missing BMI had cementless or hybrid fixation compared to those with BMI reported. Given the NJR annual report suggests poorer implant survival in cementless TKR [3], this difference could result in reduced survival overall and depending on how BMI is distributed among high-BMI patients could bias our results either way, although these fixation methods are only used in a small proportion of patients (4.1% of those with complete data and 7.6% of those with incomplete data). Overweight and obese patients receiving the operation are probably healthier and fitter than similar people not having surgery, which is likely to result in selection bias. As with all registry data, analyses are only as good as data entered; the first NJR data quality audit suggested that 95.7% of primary TKRs and 90.3% of revision TKRs were captured in financial year 204/15. Despite this high level of completeness, at the time of data collection, the NJR did not routinely capture operations where implants were not added, removed, or modified. This means that if a patient returned to theatre for an operation that did not involve the change of any implants, it would not have been captured by the NJR and would therefore not be reported by our study. It is possible that patients may require revision surgery but are deemed unsuitable because of comorbidities, and, as such, are not identified by the NJR as a failure. While this is a recognised limitation of registry research, it may be particularly relevant in this study if patients with high BMI at the time of primary surgery are considered at higher risk of developing future comorbidities that would render them less fit for revision surgery. OKS data were only available up to 6 months after TKR so we were unable to assess patient-reported pain and function as long postoperatively as we could describe revision outcomes. It is possible that recovery trajectories could vary according to BMI (i.e., higher BMI patients taking longer to recover). This could mean that patients in one particular group may not have

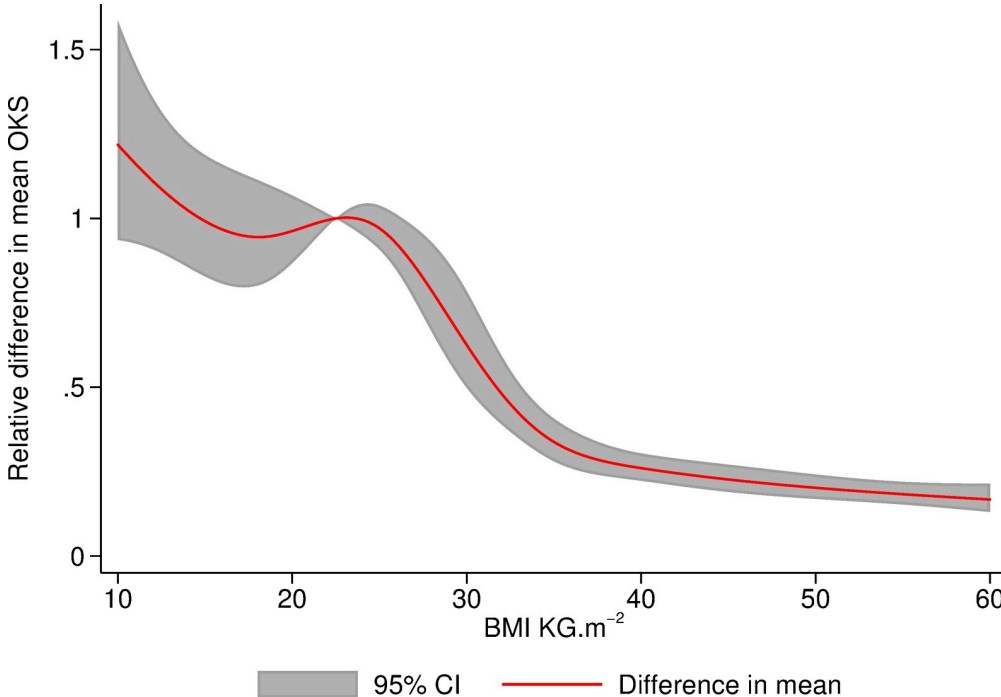

**Fig 6. Estimates of the postoperative score in relation to the preoperative OKS by BMI category.** We extracted these estimates from the fully adjusted model described in Table 7. BMI, body mass index; OKS, Oxford Knee Score.

achieved their peak postoperative outcome score by the 6-month point reported in this study. While the OKS has been widely validated, it has not been specifically validated in a solely high BMI group. This could potentially create some bias in comparison of subgroups of BMI if those with high BMI are more likely to score certain questions either higher or lower than patients with normal BMI. This study is observational in nature, and, as such, statements about causality cannot be made. Data used are routine data, and, as such, not collected specifically for inclusion in this study; this may lead to misclassification of covariates, missing data, and residual confounding.

## Strengths and weaknesses of the study in relation to other studies

The results described here conflict with previously observed associations of higher BMI with increased all-cause mortality in general nonsurgical populations [27]. This may reflect a healthy surgery effect (obesity paradox), where those with high BMI selected for surgery are fitter with fewer comorbidities than those who do not present themselves, or are deemed unsuitable, for surgery. Our observation that mortality rates following primary TKR were similar or lower at high BMI is consistent with some previous studies [28]. A U-shaped relationship between BMI and mortality has been noted in 2 studies with higher mortality in underweight patients (BMI $<18.5$ kg/m$^2$) compared to patients with a "normal" BMI according to WHO criteria [19,29]. Individual units or surgeons may employ different methods of determining a patient's fitness for surgery as well as differing pre- and postoperative care for these patients. The data available in our study did not allow this to be explored in more depth. Our results do suggest that the processes already in place are suitable in identifying those high-BMI patients at increased risk of death and that restricting access to surgery at the point of referral is unlikely to be of benefit. In an analysis of data from over 54,000 patients undergoing primary

TKR in the UK, there was a 1.02% increased hazard of revision for each unit of BMI, which is consistent with our study [30]. In a systematic review and meta-analysis including studies of primary TKR reported before February 2017, Pozzobon and colleagues note that in 5 studies, long-term pain, and, in 10 studies, disability, were greater in patients with BMI $\geq$30 kg/m$^2$ compared with BMI <30 kg/m$^2$ [31]. Due to the use of different outcome measures, the authors did not report whether these outcomes were clinically relevant. Our findings are generally conflicting with those of Chaudhry and colleagues, who in 2019 published a meta-analysis suggesting higher risk of revision and worse patient-reported outcomes in "severely, morbidly and super-obese patients" [32]. The main limitation of their analyses was the quality of included studies. Their conclusions focused on revision rate being driven by septic revisions, a subgroup we did not specifically look at in our study. Similarly, to Chaudhry and colleagues, we reported an increased revision risk in patients with higher BMI but concluded the cumulative revision estimate was still below the nationally recognised benchmark.

### Meaning of the study: Possible explanations and implications for clinicians and policy makers

The results of this study are important for patients, surgeons, and healthcare commissioners, in that patients with a high BMI do not appear to have clinically relevant poorer outcomes compared to those with "normal" BMI. This is particularly relevant given the large absolute numbers of obese patients (273,565; 55.4%) that have received surgery and the incidence of symptomatic knee osteoarthritis and its progression increases with BMI [33]. Regardless of the observed differences in the 10-year cumulative revision estimates between groups, these estimates are all still comfortably within the nationally recognised benchmark of 5% at 10 years. Patients with higher than "normal" BMI showed smaller relative improvements in pain and function scores at 6 months after TKR, but this is outweighed by substantial improvements across all BMI categories. Improvements in OKS across categories ranged from 18 to 20 points, consistent with patient reporting of knee problems being "much better" than before surgery, and the difference between groups was lower than the clinically relevant difference of 4/48 reported by Beard and colleagues [21]. It is important to emphasise that, although we have detected statistically significant differences due to the very large sample size, they are not clinically meaningful differences.

### Unanswered questions and future research

The main unanswered question from this work is what the OKS of patients will be at longer follow-up intervals, but these data are not yet available. The "healthy patient effect" that we propose in the setting of TKR for patients with higher than "normal" BMI also warrants further investigation. Patients with high BMI in combination with other risk factors (such as comorbidity) may have filtered out naturally in our cohort, suggesting that additional BMI-based filters are not needed at the referral stage. We did not investigate factors such as length of stay, which may have an impact on cost-effectiveness or primary TKR in this study. If BMI changes length of stay, it may lead to increased costs; therefore, future studies could investigate the effect of cost-effectiveness as an outcome. If it is accepted that BMI is not an appropriate rationing tool for TKR, then work looking at whether other instruments such as preoperative OKS assessments could be used may be useful.

### Conclusions

In this study, revision, mortality, and pain and functional outcomes in obese patients appear to be similar to patients with a "normal" BMI at the time of surgery. Limiting access to TKR based on BMI thus appears to be unfounded.

## Supporting information

**S1 Checklist. STROBE and RECORD checklist.** RECORD, Reporting of studies Conducted using Observational Routinely-collected Data; STROBE, Strengthening the Reporting of Observational Studies in Epidemiology.
(DOCX)

**S1 Fig. Flowchart diagram describing the steps to create dataset for PROMs analyses.** PROMs, Patient Reported Outcome Measures.
(TIF)

**S2 Fig. Cumulative probability of revision (KM estimates) with at risk table by BMI category.** BMI, body mass index; KM, Kaplan–Meier.
(TIF)

**S3 Fig. Cumulative probability of 90-day mortality (KM estimates) with risk table by BMI category.** BMI, body mass index; KM, Kaplan–Meier.
(TIF)

**S1 Table. Estimates from the flexible parametric survival model to investigate the association of BMI and revision using the NJR–HES dataset.** All the models were fitted on the hazard scale using 4 degrees of freedom. Adjusted models adjust for age, gender, type of fixation, ASA grade, year of having the primary operation, indication for operation, IMD, and Charlson comorbidity index. ASA, American Society of Anaesthesiologists; BMI, body mass index; HES, Hospital Episodes Statistics; IMD, Index of Multiple Deprivation; NJR, National Joint Registry; TKR, total knee replacement.
(DOCX)

**S2 Table. Estimates from the Cox regression model to investigate the association of BMI and mortality within 90 days using the NJR–HES dataset.** Adjusted models adjusted for age, gender, ASA grade, year of primary TKR, indication for operation, IMD, and Charlson comorbidity index. ASA, American Society of Anaesthesiologists; BMI, body mass index; HES, Hospital Episodes Statistics; IMD, Index of Multiple Deprivation; NJR, National Joint Registry; TKR, total knee replacement.
(DOCX)

**S3 Table. Coefficients of BMI categories to predict the mean increase or decrease on the postoperative OKS after 6 months.** Adjusted model adjusts for age, gender, ASA grade, indication for operation, fixation type, and year of receiving the primary TKR, anxiety status, Charlson score, and multiple deprivation index. ASA, American Society of Anaesthesiologists; BMI, body mass index; OKS, Oxford Knee Score; TKR, total knee replacement.
(DOCX)

## Acknowledgments

We thank the patients and staff of all the hospitals who have contributed data to the National Joint Registry (NJR). We are grateful to the Healthcare Quality Improvement Partnership (HQIP), the National Joint Registry Steering Committee (NJRSC), and staff at the NJR Centre for facilitating this work.

**Disclaimers:** The views expressed represent those of the authors and do not necessarily reflect those of the NHS, the National Institute for Health Research, the Department of Health, NJRSC, or HQIP who do not vouch for how the information is presented.

## Author Contributions

**Conceptualization:** Sofia Mouchti, Ashley William Blom, Jeremy Mark Wilkinson, Michael Richard Whitehouse, Andrew Judge.

**Data curation:** Sofia Mouchti, Jeremy Mark Wilkinson, Michael Richard Whitehouse, Andrew Judge.

**Formal analysis:** Jonathan Thomas Evans, Sofia Mouchti, Andrew Beswick, Andrew Judge.

**Funding acquisition:** Ashley William Blom, Michael Richard Whitehouse.

**Investigation:** Jonathan Thomas Evans, Sofia Mouchti, Ashley William Blom, Jeremy Mark Wilkinson, Michael Richard Whitehouse, Andrew Beswick, Andrew Judge.

**Methodology:** Jonathan Thomas Evans, Sofia Mouchti, Ashley William Blom, Jeremy Mark Wilkinson, Michael Richard Whitehouse, Andrew Beswick, Andrew Judge.

**Project administration:** Andrew Judge.

**Supervision:** Ashley William Blom, Jeremy Mark Wilkinson, Michael Richard Whitehouse, Andrew Beswick, Andrew Judge.

**Validation:** Jonathan Thomas Evans.

**Visualization:** Jonathan Thomas Evans, Andrew Judge.

**Writing – original draft:** Jonathan Thomas Evans, Andrew Beswick, Andrew Judge.

**Writing – review & editing:** Jonathan Thomas Evans, Ashley William Blom, Jeremy Mark Wilkinson, Michael Richard Whitehouse, Andrew Beswick, Andrew Judge.

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
