## [Editor Report · Decision Letter 0]

23 Sep 2020

Dear Dr Evans, 

Thank you for submitting your manuscript entitled "The effect of obesity on revision surgery, mortality and patient reported outcomes of primary knee replacement surgery: data from the National Joint Registry for England, Wales, Northern Ireland and the Isle of Man" for consideration by PLOS Medicine.

Your manuscript has now been evaluated by the PLOS Medicine editorial staff [as well as by an academic editor with relevant expertise] and I am writing to let you know that we would like to send your submission out for external peer review.

Kind regards,

Helen Howard, for Clare Stone PhD 

Acting Editor-in-Chief

PLOS Medicine 

plosmedicine.org

---

## [Decision Letter · Decision Letter 1]

13 Nov 2020

Dear Dr. Evans,

Thank you very much for submitting your manuscript "The effect of obesity on revision surgery, mortality and patient reported outcomes of primary knee replacement surgery: data from the National Joint Registry for England, Wales, Northern Ireland and the Isle of Man" (PMEDICINE-D-20-04558R1) for consideration at PLOS Medicine. 

Your paper was evaluated by a senior editor and discussed among all the editors here. It was also evaluated by three independent reviewers, including a statistical reviewer (r#1). The reviews are appended at the bottom of this email and any accompanying reviewer attachments can be seen via the link below:

[LINK]

In light of these reviews, I am afraid that we will not be able to accept the manuscript for publication in the journal in its current form, but we would like to consider a revised version that addresses the reviewers' and editors' comments. Obviously we cannot make any decision about publication until we have seen the revised manuscript and your response, and we plan to seek re-review by one or more of the reviewers. 

We expect to receive your revised manuscript by Dec 04 2020 11:59PM. Please email us (plosmedicine@plos.org) if you have any questions or concerns.

We look forward to receiving your revised manuscript. 

Sincerely,

Emma Veitch, PhD

PLOS Medicine

On behalf of Adya Misra PhD, Senior Editor, 

PLOS Medicine

plosmedicine.org

*Please structure your abstract using the PLOS Medicine headings (Background, Methods and Findings, Conclusions); we also recommend authors include a brief note about any key limitation(s) of the study's methodology in the abstract, ideally in Methods and Findings subsection.

*At this stage, we ask that you include a short, non-technical Author Summary of your research to make findings accessible to a wide audience that includes both scientists and non-scientists. The Author Summary should immediately follow the Abstract in your revised manuscript. This text is subject to editorial change and should be distinct from the scientific abstract. Please see our author guidelines for more information: https://journals.plos.org/plosmedicine/s/revising-your-manuscript#loc-author-summary

*If possible please reformat the citation style into PLOS Medicine's format (should be straight forward if using referencing software) - this should use callouts formatted as sequential numerals in square brackets (not superscript).

*The authors might wish to look at established guidance for reporting studies conducted using routinely-collected datasets (which include some items related to record linkage between databases) - eg the RECORD guidance - https://www.equator-network.org/reporting-guidelines/record/;
https://journals.plos.org/plosmedicine/article?id=10.1371/journal.pmed.1001885. If using that statement to report the study we'd suggest the authors note this in the Methods, citing the RECORD paper, and then also append as a supplementary file the completed RECORD checklist.

*Can the authors clarify in the paper (methods section) whether the analysis reported here corresponded to one laid out in a prospective protocol or analysis plan? Please state this (either way) early in the Methods section.

*In Discussion, "unanswered questions...", the following sentence seems to be lacking an additional phrase, see **xx** below. 

"If it is accepted that BMI is not an appropriate rationing tool for TKR, then work looking at whether other instruments such as pre-operative OKS **assessments could be used**, may be useful.

Comments from the reviewers:

Reviewer #1: 

I confine my remarks to statistical aspects of this paper.

The general approach is fine and I like the use of new methods and the fact that the authors included all the covariates rather than doing some sort of model building.

However, I have a few issues to resolve before I can recommend publication.

The main issue is that BMI should not be categorized for the analysis (although the categories may be useful for some parts of presentation). Leave BMI as a number and investigate nonlinearity with a spline. The use of a spline on a categorized version of a variable is very limited. It limits estimation of knots - they have to happen at the cutoffs between categories. See *Regression Modeling Strategies* by Frank Harrell for more problems with categorizing continuous variables.

Grouping age has some of the same issue, but the authors use so many groups that it's probably OK. Still, why not leave age as "years"? 

Why was year grouped the way it was? That would seem to force year to be categorical and would waste degrees of freedom (not really a big problem here) and make interpretation of any year effect harder. 

Fig 2 The y axis should be rescaled to soimething like 0 to 5 or 6. As is, most of the graph is blank. This makes it harder to see the rela

Fig. 3. I don't like that the scales of the two axes are so different. The plot makes it look like there was equal improvement for all initial conditions, when, really, there was no improvement at the highest initial leves. I suggest either making both axes go from 0 to 50 or else a graph with initial score on the x axis and improvement on the y axis. This seems very important. The authors note that BMI isn't a useful predictor of success (and, as an overweight man who will probably need knee surgery, I appreciate that!) but another useful finiding here (which might make another paper) is that people with good initial OKS didn't really improve much.

Peter Flom

Reviewer #2: 

The authors have written a clear and logically constructed article suggesting the higher body mass index is associated with slightly higher revision rate but similar clinical improvement (in terms of Oxford Knee Score) after six months after primary knee replacement. The case numbers are clearly sufficient and the follow-up long enough to report revision rates. Nationally representative materials and appropriate statistical methodology can be also considered as strengths of this study. Unfortunately, there are also several issues to criticize. However, I believe the authors have good possibilities to satisfactorily respond to these concerns.

Major concerns:

1) I have troubles following the description of collection of the data and creating the datasets. I understand that revision data and clinical data come from different sources and are from different time periods, but presenting four (!) different flowcharts is confusing. Figure 1 of the manuscript is quite ok although I do not understand why to report the total number of TKRs since 2003 although included cases were from 2005-2016. I am also able to follow Figure 1 in the supplementary materials but do not understand the next flowchart Figure 2 at all (it ends up with figure 288,423 that is not mentioned anywhere else) and the last flowchart feels like repetition. Also, the different time periods are confusing and poorly explained acronyms (LM, HES) complicate interpretation. The authors should bear in mind that most readers are likely not familiar with the registers of the authors' country.

It is noteworthy that there are lots of missing data. About one third of cases were excluded because of missing BMI and there were exclusions for other reasons as well. I do not quite agree with the authors that excluded patients were comparable to the included ones. According to table 1, there were more patients with lower ASA and more cementless TKRs in the incomplete (i.e. excluded) group. Differences e.g. in comorbidity, type of operating unit and social status are not reported. It is also likely that patients responding to Oxford Knee Score represent a selected sample of patients (younger, uncomplicated and so on), limiting generalizability of the findings). These issues related to possible selection bias should be reported and discussed more thoroughly. Especially the results about lower mortality with higher BMIs indicate that in the obese groups there have been significant patient selection (and/or differences in pre- and perioperative care) that are not fully captured by the variables used in this study.

2) Validity and coverage of the data sources should be described. Although national registers usually cover primary joint replacements quite well, the coverage of revision joint replacements is usually poorer. From the viewpoint of the current study, prosthetic-joint infections represent a specific problem as their risk is higher in patients with highest BMI, and they can be managed with other means than full revision surgery especially in high-risk patients and in acute infections. One might have expected higher early revision rate in the most obese group because of PJIs. It is unclear how data about different confounding variables were collected, what is their completeness and coverage. Most readers may not be familiar with Multiple Deprivation Index and EQ5D, the coverage of latter of which could be expected to be limited.

3) The materials were collected until December 31, 2016, and the same date was the end of follow-up. Consequently, some of the patients had practically zero follow-up and they could not have experienced the outcomes that were of interest. Although for example in the analysis of deaths, Cox analysis (taking into account different follow-up times) was used appropriately, including also patients without sufficient follow-up in the denominator leads to falsely low absolute rates of e.g. deaths or revisions. Now that it is already year 2020, it is difficult to see, why the authors have not used more later data - or in the other hand, made the choice of materials so that all patients would have reasonable minimum follow-up (at least six months to get PROM data or more preferably one year).

There are also discrepancies in the observation period: methods section say that cases were included from 2005-2016, but flowcharts in the supplementary materials and results section also talk about period 2009-2014. 

The adjustments used in the regression models are also described incompletely and/or contradictory: methods section lists several confounding factors but not all of them were used in all analyses, according to legends of 4-6 and the tables presented in supplemental materials. Most materials of supplementary materials are not referred to in the text. As there have been several multivariable models, statements like "fully adjusted model" are not accurate enough.

4) I'd appreciate more critical analysis and interpretation of the results and more thorough review of the literature.

- With all due respect, now the manuscript has a feeling that its aim is to support the view that BMI should not have a role in rationing knee replacement although earlier literature and also the current results might be interpreted to support the opposite view. There are a couple of recent meta-analyses (indicating poorer results with higher BMI) that have been ignored (Chaudhry et al. JBJS Rev 2019, Sun & Li Knee 2017, Si et al. Knee Surg Sports Traumatol Arthrosc 2015) - and considering numerous earlier reports, it should be articulated more clearly what are the news that the present study give (mid-term revision rates could be such). The present study is somewhat contradictory with earlier meta-analyses and reasons for these discrepancies should be commented.

- Early complications (not leading to revision joint replacement), possibly prolonged hospitalizations and particularly prosthetic-joint infections have not been taken into account in the present analysis, which should be discussed as a limitation as they affect the costs and hence cost-effectiveness and rationing of surgery. Also one further reason supporting conservative treatment rather than surgery is that weight loss and rehabilitation have been reported to ease the symptoms of osteoarthritis, making knee replacement possibly unnecessary at least for a while.

- The authors suggest that reporting revision rates, mortality and clinical outcome all in the same study is a strength of this study. Combining the three but analyzing them separately however does not necessary give any extra over separate and possibly more detailed studies. In the Discussion, the authors write "The failings of examining single domains have previously been highlighted, in that just because a TKR has not been revised does not necessarily mean it was a success." to reason combining the three outcomes. Because the present study analyzes clinical outcome after six months but revision rates up to 10 years, it does not overcome this problem, and problem with falsely low revision rate as a result of not performing revision in hight risk patients still remains a problem and should be acknowledged.

Minor concerns:

- References 8 and 9 are about hip replacements although there are suitable ones about knees as well.

- Age is a continuous variables. Why is it used as classified one?

- How (staged or same-day) bilateral knee replacements were dealt with? This is relevant both concerning the outcome analysis but also data linkage.

- Fulfillment of proportional hazards assumption should be commented in association with use of Cox regression analysis.

- Please, keep the order of analyzed outcomes the same throughout the manuscript (now OKS is mentioned first in the results section).

- What does BMI <0 mean?

- Y-axis in Figure 3 should start from zero or it should be indicated that it is cut.

- It seems that Figure 3 represents data from a linear model. In its current form its value is questinable and interpretation is difficult: I think it is not that certain that association is linear but instead from lower scores the improvent might be greater. Ceiling effect may also be present. Secondly, there is possibly variation in individual patients' responses and one might hypothesize that variation might be greater in the obese groups. I'd suggest replacing Figure 3 with information based on actual observations (change from pre to postop in different BMI categories and its variation). 

- I's suggest using Kaplan-Meier analysis (instead of modelled curves of Figure 2) in the manuscript as results from adjusted models are already shown in the tables.

- Although the change in OKS in different BMI groups is within minimal clinically meaningful difference, it is noteworthy that the differences in absolute OKSs are greater than 4 and remain essentially unchanged after operation.

- 90-day mortality is reported as the outcome of interest, but also 30-day and 60-day mortality are reported in Table 3. Please, select which are the outcomes and at which time points they are analyzed. As it comes to mortality, I understand that 90-day mortality can be considered to represent the risk related to surgery. When weighing the value of surgery, also mortality in longer term is relevant: the benefit of surgery could be interpreted different e.g. if the most obese patients had high 5/10-year mortality. Furthermore, if there are significant differendes in longer-term mortality, it may bias K-M and Cox analyses and one might argue that compering risk analyses should be used instead.

- I do not consider reporting p values necessary when HRs are shown with 95% CI.

- Please replace p=0.0000 with p<0.001 as the p(probability) is hardly zero.

- Supplemental figure 1: There are 252,659 PROMs-HES with unique NJR index numbers, of which ca. 15000 disappear after linkage (there are just 237,288 at the final step). This is not explained or illustrated satisfactorily (a box indicating the faith of these 15000 cases should be added).

Reviewer #3: 

This is a very interesting study of outcomes of TKA for obese patients. It is highly relevant for all the reasons highlighted by the authors. Also outside the UK, we see financially motivated attempts to reduce the number of operations, some of which appear entirely unjustified, e.g. requirements for months of physiotherapy in cases of OA with severe deformity, or refusing private cover for patients with a BMI over 35.

A register is not able to provide answers on causative relations, as the authors correctly argue, and the true effect of a factor on the outcome of a procedure is probably beyond the ability of registers. They are, however, one of the best means of getting information on current practice and current relations between possibly causative (and possibly confounded) factors and outcome. And no better starting point than the largest register.

The results are important, and the consequences of this study are potentially wide-reaching with general effects on health policy.

Does the NJR have information on postoperative weight (and weight loss)? If so, it would be very interesting to have the information included in the analyses.

Introduction

* Very good introduction. Although the aim appears well described, there is some uncertainty, that influences the judgement of appropriateness of the methods used. For instance: is there any time frame for the PRO outcome? Does the study aim to determine PRO improvements in the recovery period or the plateau outcome? Or is this a pragmatic analysis of the available data?

Methods

* Why were patients excluded, if the indication for revision was unknown? Revision is revision, and since indications may be less reliable, I would include all types of revision.

* Why was the analysis restricted to TKR? No interest in unicompartmentals?

* The definition of revision is the well-known triad of removal, addition or exchange. Since a higher infection in obese patients has been suggested, and since some early infections may result in a reoperation not qualifying for a revision, some early complications may have been missed. This should be discussed. There may well be a general need to change focus from revisions to all types of reoperations, even if this deviates from tradition.

* It is not correct to suggest that the OKS covers 12 domains. The questionnaire was developed as a one-domain questionnaire (knee function and pain, not clearly separated), and it resulted in 12 questions within that domain.

* Had the authors had long-time PRO-data, one might argue for not excluding patients after revision. With a 6-month PRO time-frame, the exclusion appears justified.

* The reference to Royston and Parmar should be listed in the references. I am not an expert on the model used, and I would suggest a statistical review of the appropriateness of the model.

* Is BMI data entered directly into the NJR, or is BMI calculated from entries of bodyweight and height? I would expect more errors with the former entry type. Please specify.

* Is there no anchor question in the NJR? Would be very interesting to see this in relation to the OKS improvements instead of a simple reporting of the crude OKS improvements. See comment under Discussion below.

Results

* Comparing the complete and incomplete datasets, the ASA distributions do differ significantly, although possibly similar(?). The same is the case for fixation type. Possible reasons? Consequences? Comments?

* Last 3 lines of Revision paragraph: It should be emphasized, that the 30-35 kg/m2 group did not differ significantly from the normal BMI group. To the casual reader, the message is an 8% higher revision risk, which may not be true.

* First paragraph on OKS: This is just a repetition of table 6. Please provide an interpretation of the data.

Discussion

* A possible limitation is the use of the OKS. The instrument was developed for knee OA patients, but as far as I remember, there was no mentioning of BMI for the focus group patients in the development paper. I also do not remember a validation paper for obese patients. The importance of this is that obese patients may have other needs and expectations, and the quantification of their feelings and function may well be biased by using the same instrument for both groups. As an example, I would assume that obese patients would have far more difficulty in kneeling (question 7) than non-obese patients, but this does not necessarily imply a problem with the surgical outcome. This should be discussed.

* The timing of the post-operative OKS is a certain limitation that should be discussed. By comparing the two groups of patients at a single and identical time point assumes knowledge of identical improvement curves for the groups. It may well be that obese patients recover more slowly than non-obese patients, and the 6-month comparison may give a biased estimate of the "true" difference in outcome between the groups. One group may be measured in the recovery phase, while the other is measured in the plateau phase. This should also be discussed.

Figures

1. It hardly seems necessary to specify reasons for exclusion of no patients.

2. Y-axis should be rescaled to better visualize differences between curves, e.g. range 0-5%.

3. A graphical representation of the uncertainty of the estimate curves would increase the value of this figure. It would be difficult to represent all five, but a cumulated model could be shown.

Nov 10, 2020

Anders Odgaard

Rigshospitalet, Copenhagen University Hospital

[LINK]

---

## [Decision Letter · Decision Letter 2]

12 Jun 2021

Dear Dr. Evans,

Thank you very much for re-submitting your manuscript "The effect of obesity on revision surgery, mortality and patient reported outcomes of primary knee replacement surgery: data from the National Joint Registry for England, Wales, Northern Ireland and the Isle of Man" (PMEDICINE-D-20-04558R2) for consideration at PLOS Medicine.

I have discussed the paper with editorial colleagues and it was also seen again by two reviewers. I am pleased to tell you that, provided the remaining editorial and production issues are fully dealt with, we expect to be able to accept the paper for publication in the journal.

[LINK]

Please let me know if you have any questions in the meantime, and we look forward to receiving the revised manuscript.   

Sincerely,

Richard Turner, PhD

rturner@plos.org

Requests from Editors:

Bearing in mind the observational research design, we ask you to amend the title to better accord with journal style, and suggest: "Obesity and revision surgery, mortality and patient reported outcomes after primary 

knee replacement surgery in the National Joint Registry for England, Wales, Northern 

Ireland and the Isle of Man: A cohort study".

Please avoid words such as "effect", implying causal inference, throughout the text unless supported by evidence.

In the abstract, to "After adjustment ...", please add a summary of what has been adjusted for.

In the abstract and main text, please quote p values alongside 95% CI, where available. 

Please attach the analysis plan as a supplementary document, referred to in your Methods section. 

We did not find the RECORD checklist attached - please include this with your revision. 

In the checklist, please ensure that individual items are referred to by section (e.g., "Methods") and paragraph number, not by line or page numbers as these generally change in the event of publication. 

Please remove the "Role of the funding source" section from the Methods. 

Please remove information on funding and competing interests from the end of the main text. In the event of publication, this information will appear in the article metadata, via entries in the submission form. 

Throughout the article, please quote exact p values or, for smaller values, p<0.001.

Throughout the text, please style reference call-outs as follows: " ... most patients [1,2]." (noting the space preceding the square bracket and the absence of spaces within the brackets). 

Please restrict "world's largest joint registry" to one mention, and add "to our knowledge", or similar.

In the reference list, please use journal name abbreviations (e.g., "Lancet") consistently.

Comments from Reviewers:

*** Reviewer #1: 

The authors have addressed my concerns and I now recommend publication

Peter Flom

*** Reviewer #2: 

The authors have done thorough work in revising the manuscript according to my and other reviewers' comments. Most importantly, the manuscript is much easier to follow now. 

There are just a couple of issues that I'd suggest considering.

1) Supplementary data (S2) now shows number of cases that are still under follow-up at different timepoints. I think this is important information and do not see a reason to "hide" it in the supplementary data. I'd suggest adding the number of cases under follow-up into Figure 2. I may not be necessary to show the numbers for each BMI category. Just the total number could be enough.

2) I think it would be important to show mortality during the follow-up years. In the current analyses regarding prosthesis survival, closure of follow-up and death are dealt with similarly (censoring of the case). As I already commented, mortality is likely different in different BMI groups (and similarly the type of ending the follow-up: closure of follow-up, revision, death) in long-term follow-up and from the viewpoint of considering risks and benefits of surgery, such differences are clinically relevant. 

3) Figure 5 shows linear association between preoperative and postoperative OKS scores. The authors have only partly answered to my previous comment. The parts that remained unanswered are: "In its current form its value is questionable and interpretation is difficult: I think it is not that certain that association is linear but instead from lower scores the improvement might be greater." Secondly, there is possibly variation in individual patients' responses and one might hypothesize that variation might be greater in the obese groups. I'd suggest replacing Figure 3 with information based on actual observations (change from pre to postop in different BMI categories and its variation)." I'd like to emphasize that also variation might be of interest. In it's current form, I do not see much value for the current Figure 5.

***

[LINK]

---

## [Editor Report · Decision Letter 3]

21 Jun 2021

Dear Dr Evans, 

On behalf of my colleagues and the Academic Editor, Dr Jamsen, I am pleased to inform you that we have agreed to publish your manuscript "Obesity and revision surgery, mortality and patient reported outcomes after primary knee replacement surgery in the National Joint Registry: A cohort study" (PMEDICINE-D-20-04558R3) in PLOS Medicine.

Prior to final acceptance, we suggest adding "UK" to the title; please also remove the information on funding from the abstract; and convert "p<0.0001" to "p<0.001" in the supplementary tables.

PRESS

Sincerely, 

Richard Turner, PhD 

rturner@plos.org